# T Lymphocytes and Their Potential Role in Dementia with Lewy Bodies

**DOI:** 10.3390/cells12182283

**Published:** 2023-09-15

**Authors:** Jay Amin, Claire Gee, Kiran Stowell, Daisy Coulthard, Delphine Boche

**Affiliations:** 1Clinical Neurosciences, Clinical and Experimental Sciences, Faculty of Medicine, University of Southampton, Southampton SO17 1BJ, UK; 2Memory Assessment and Research Centre, Tom Rudd Unit, Moorgreen Hospital, Southern Health NHS Foundation Trust, Southampton SO30 3JB, UK

**Keywords:** dementia with Lewy bodies, inflammation, adaptive immunity, lymphocytes, T cells

## Abstract

Dementia with Lewy bodies (DLB) is the second most common neurodegenerative cause of dementia. People with DLB have an inferior prognosis compared to Alzheimer’s disease (AD), but the diseases overlap in their neuropathology and clinical syndrome. It is imperative that we enhance our understanding of the aetiology and pathogenesis of DLB. The impact of peripheral inflammation on the brain in dementia has been increasingly explored in recent years, with T lymphocyte recruitment into brain parenchyma identified in AD and Parkinson’s disease. There is now a growing range of literature emerging on the potential role of innate and adaptive immune cells in DLB, including T lymphocytes. In this review, we examine the profile of T lymphocytes in DLB, focusing on studies of post-mortem brain tissue, cerebrospinal fluid, and the blood compartment. We present an integrated viewpoint on the results of these studies by proposing how changes to the T lymphocyte profile in the brain and periphery may relate to each other. Improving our understanding of T lymphocytes in DLB has the potential to guide the development of disease-modifying treatments.

## 1. Introduction

Dementia with Lewy bodies (DLB) accounts for 5–10% of all dementia cases, and it is the second most common degenerative cause of dementia after Alzheimer’s disease (AD) [1]. Clinically, DLB is characterised by fluctuating cognition, recurrent visual hallucinations, motor features of parkinsonism, and rapid eye movement sleep behaviour disorder [2]. Compared to AD patients, people with DLB experience increased hospitalisation, accelerated cognitive decline, and a shorter lifespan [3,4]. There is a critical need to enhance our understanding of the pathophysiology of DLB, enabling identification of targets for drug discovery.

DLB is defined pathologically by the presence of alpha-synuclein (α-syn) aggregates in the cerebral cortex and basal ganglia, which manifest as Lewy bodies and/or Lewy neurites, termed Lewy-related pathology (LRP) [5,6]. Associated pathological features of DLB include neuronal and synaptic loss and neurotransmitter deficits [5]. There is an extensive overlap between DLB and Parkinson’s disease (PD), with nigrostriatal LRP accumulation and dopaminergic neuronal degeneration being the characteristic pathological findings in PD [7]. Indeed, dementia in Parkinson’s disease (PDD) is an almost inevitable part of the prognosis of PD [8]. An overlap in neuropathology also exists between DLB and AD, with amyloid-beta (Aβ) plaques and hyperphosphorylated tau (P-tau) tangles contributing to accelerated cognitive decline in DLB [5,9]. DLB and PDD share similar clinical features and are distinguishable by the timing of the onset of motor symptoms compared to cognitive decline [10]. DLB is diagnosed when dementia develops prior to, or within one year of the onset of, parkinsonism, whereas PDD is diagnosed when dementia develops at least one year after the onset of parkinsonism [2].

Neuroinflammation is increasingly recognised as a key factor in the pathogenesis and progression of neurodegenerative diseases and has been widely explored in AD [11]. Microglia, the resident immune cells of the central nervous system (CNS) and part of the innate immune system, play an important role in responding to neuropathology in AD [12]. These cells become chronically activated with a deleterious impact on their function in the AD brain [13,14]. In the periphery, chronic inflammatory conditions, such as diabetes mellitus and obesity, have been shown to increase the risk of late-onset AD, possibly through inducement of neuroinflammation [11]. This is accompanied by alterations in peripheral blood cell populations, including increased numbers of neutrophils and decreased numbers of lymphocytes (T and B cells) [15]. Neutrophils can increase in response to raised inflammatory cytokine levels, which have been implicated in the pathogenesis of AD [16]. A decrease in peripheral T cells raises the possibility that a proportion or subpopulation of T cells could be migrating into the brain. Indeed, preclinical and human post-mortem studies in AD have consistently demonstrated the presence of T cells in brain parenchyma [17,18,19,20].

Reduced B cell counts in the periphery are associated with normal ageing, although a more prominent reduction is observed in AD. In comparison to AD, the role of inflammation in DLB has been less extensively researched. However, there is now growing evidence to suggest alterations in cerebral and peripheral immune profiles in DLB [21]. It has been proposed that α-syn may activate the innate and adaptive immune responses in DLB [22], and extracellular α-syn aggregates have been shown to activate microglia and astrocytes via toll-like receptors in the brain to produce pro-inflammatory cytokines [23]. Furthermore, aggregation of α-syn in PD post-mortem brain tissue and preclinical models has been associated with neuroinflammation, reactive microgliosis, and T cell infiltration [24,25]. Data from human studies examining blood cytokines and positron emission tomography (PET) imaging using the inflammatory marker TSPO show an increase in inflammation in early DLB, which then decreases with disease progression. Although post-mortem human brain studies have failed to demonstrate significant microglial activation in DLB, increased cortical recruitment of T cells has indicated a potential role for adaptive immunity [21,26]. The presence of T cells in the AD and PD brain supports the possibility of interaction between the cerebral innate immune system and the peripheral adaptive immune system, and it has been proposed that T cell infiltration into the brain promotes crosstalk between T cells and microglia, resulting in the acceleration of neuroinflammation [27].

Our understanding of the role of T cells and the wider adaptive immune system in DLB remains limited. Here, we present a summary of the physiological role of T cells, followed by evidence of their involvement in DLB, exploring alterations in the profile of T cells in the brain, cerebrospinal fluid (CSF), and blood compartment.

## 2. Physiological Role of T Cells

In contrast to the swift and non-specific approach of the innate immune system, the adaptive immune system provides a targeted immune response that functions to eliminate pathogen-infected cells and to develop immunological memory [28]. The adaptive immune system comprises T and B cells, antigen-presenting cells (APCs), such as macrophages, and antibodies [28]. T cells are broadly classified into helper (CD4+) and cytotoxic (CD8+) T cells. The activation of T cells requires specific recognition between T cell receptors (TCRs) and foreign antigen peptides presented by major histocompatibility complex (MHC) molecules. CD4+ T cells interact with APCs via MHC class II (MHCII), resulting in T cell proliferation and cytokine release. There are several subtypes of CD4+ T cells, determined by the cytokines secreted, which are involved in various immunological processes, such as B cell maturation and macrophage activation [29]. The CD4+ T cell subset functions and effector cytokines are summarised in Table 1.

CD8+ T cells contribute to the clearance of intracellular pathogens and neoplastic cells. Once activated by the recognition of antigen on MHC Class I (MHCI) molecules, they undergo differentiation and proliferation to generate an expanded population of effector and memory T cell types. APCs and CD4+ T cells can also mediate CD8+ T cell activation through costimulatory signals and the production of cytokines [30]. An overview of CD8+ T cell subtypes and their functions is shown in Table 2.

Most nucleated cells in the body express MHCI, but only professional APCs, such as dendritic cells, B cells and macrophages, can activate CD4+ T cells via MHCII. Microglia and astrocytes expressing MHCII have been shown to directly activate CD4+ T cells, supporting their antigen-presenting capabilities [31]. Once activated, T cells can differentiate into memory cells that provide long-term immunity following re-exposure to the same antigen. Ageing is associated with a shift from naïve to more differentiated T cells, which is most evident in CD8+ subsets [32]. T cells continue to replicate with repeated exposure to pathogens but can eventually lose their ability to proliferate, reaching a stage of replicative senescence [33]. Senescent T cells, expressing elevated concentrations of inflammatory cytokines, accumulate during ageing and have been implicated in age-related diseases, such as AD [34].

The blood of healthy individuals contains more CD4+ compared to CD8+ T cells, with the loss of circulating naïve CD8+ T cells contributing to declining numbers of CD8+ T cells with increasing age [35]. The CSF contains CD4+ (and to a lesser extent CD8+) T cells, which carry out immune surveillance of the CNS [36]. Infiltration of T cells into the brain parenchyma is restricted by the blood-brain barrier (BBB), which regulates the movement of cells and solutes between the blood compartment and the CNS. Migration of immune cells across the BBB appears to be controlled by the expression of adhesion molecules, chemokines, and their receptors [37]. Reduced expression of cell adhesion molecules in CNS endothelia limits the entry of T cells and they are therefore present only sporadically in the brain parenchyma of healthy individuals [38]. During periods of neuroinflammation, the BBB can lose its integrity and become more permeable to solutes and circulating leukocytes. This is a consequence of changes in the BBB epithelium, which include increased expression of adhesion molecules, pro-inflammatory cytokines, and chemokines [39].

There are likely to be different mechanisms involved in the migration of CD4+ and CD8+ T cells into the brain parenchyma. CD8+ T cell infiltration into the brain of CL4 transgenic mice has been shown to occur only when the cognate antigen is present in the parenchyma. Recognition of an antigen presented by MHCI on the luminal surface of the cerebral endothelium provides a mechanism for CD8+ T cell migration across the BBB [40]. Due to lack of expression of MHCII by the cerebral endothelium, this mechanism does not apply to CD4+ T cells. Animal studies investigating CD4+ T cells have observed that activated T cells are able to infiltrate the brain regardless of antigen specificity [41,42,43].

## 3. T Cells in DLB

We now present a summary of evidence describing the role of T cells in DLB, exploring changes in T cell profile and distribution in post-mortem human brain tissue, CSF, and the blood compartment.

### 3.1. T Cells in the Brain Parenchyma

In the largest human post-mortem study to date investigating T cells in the middle temporal gyrus of 30 DLB and 29 control cases, an increased presence of CD3+ T cells was observed in the parenchyma in DLB. CD3+ T cells were identified more frequently in both grey and white matter, whilst no difference was observed in perivascular areas [26]. Earlier, smaller studies examining T cells in the hippocampus have reported inconsistent findings, which include similar numbers of CD3+ T cells recorded in 12 DLB cases compared to controls [17]. CD3+ T cells have been observed to be increased around blood vessels and neuropil in the hippocampus and neocortex of 8 DLB cases compared to controls. When characterised into subtypes of T cells, the majority of these cells were CD4+ T cells. A small number of CD8+ T cells were detected in the DLB hippocampus but were absent in the remaining neocortex of both DLB and control brains, suggesting that most infiltrating T cells were CD4+ T cells [22], although this needs to be confirmed in larger studies.

Infiltration of CD4+ T cells into the brain has been demonstrated in a post-mortem study examining 7 brain regions from 17 PD patients with no dementia, 11 patients with PDD, and 14 healthy controls. Increased numbers of CD4+ T cells were found in the substantia nigra in PDD compared to control cases, but there was no significant difference in the number of CD8+ T cells across the groups. The numbers of CD4+ and CD8+ T cells per mm^2^ in the amygdala were comparable between groups. However, it is important to note that there were associations between CD4+ T cells, α-syn, and human leukocyte antigen–DR (HLA–DR+)-activated microglia in this region [44], implicating the involvement of α-syn in driving neuroinflammatory responses. Notably, previous studies have shown increased numbers of HLA–DR+-activated microglia in DLB brains compared to controls [45,46]. As the HLA complex is the MHC system of humans, it can be postulated that HLA–DR+ microglia will attract and/or communicate with infiltrated T cells. However, this contrasts with findings from a large study examining markers of microglial activation in the temporal lobes of DLB and control brains, which demonstrated no differences between groups in the levels of ionised calcium-binding adapter molecule 1 (Iba1), cluster of differentiation 68 (CD68), and HLA–DR+ protein load [26]. Furthermore, no significant microglial activation was identified in DLB brain tissue using transcriptome analysis within the pulvinar [47], anterior cingulate, and dorsolateral prefrontal cortex [48]. DLB may be characterised by an absence of microglial activation, although it is unclear whether this is specific to brain region and/or stage of disease, the latter being a limitation of the human post-mortem studies.

Further studies have explored the association between α-syn aggregation and recruitment of T cells into the brain. Widespread immunoreactivity for CD3 was observed in a study of 6 DLB brains compared to an absence of CD3 expression in 4 control cases. Immunoreactivity for CD3 was closely associated with α-syn pathology in the brainstem and with neocortical LRP [49]. Higher numbers of CD3+ T cells have been observed in the substantia nigra of 2 DLB and 4 PDD brains compared to 5 controls, with a higher percentage localising to α-syn deposits and Iba1+ microglia. Interestingly, CD3+ T cells were also identified in the hippocampus, near Lewy bodies and the excitatory presynaptic marker vesicular glutamate transporter 1 (vGLUT1) [50]. Previous studies have demonstrated co-localisation of α-syn and vGLUT1+ neurons in the hippocampus of C57BL/6N mice [51,52] and it has been proposed that vGLUT1+ neurons are vulnerable to α-syn pathology, resulting in detrimental effects on synaptic transmission [53]. Increased immunoreactivity for the pro-inflammatory cytokine interleukin 17A (IL-17A) has been detected in DLB/PDD brains [50], implying that Th17 cells that express this cytokine may contribute to neuronal damage.

The role of α-syn on T cell proliferation has been investigated using a co-culture of CD4 T cells and microglia from C57BL/6 wildtype mice, isolated and pretreated with recombinant full-length-aggregated α-syn. This resulted in marked stimulation of T cell proliferation, indicating that α-syn induces MHCII expression in microglia and enhances antigen presentation, leading to a proliferation of CD4+ T cells [54]. Notably, in PD patients, peripheral CD4+ T cells have been shown to specifically react to antigenic MHCII epitopes derived from α-syn. Cytokine responses against an α-syn epitope pool were measured to enable quantification of T cell response. T cell reactivity to α-syn was significantly higher in PD patients compared to healthy controls and the majority of T cell responses were associated with interleukin 5 (IL-5) production, indicating that CD4+ T helper 2 (Th2) cells are mediating this response [55]. A further study investigating the relationship between T cell reactivity and PD pathogenesis revealed that peripheral T cell responses to α-syn were highest near to the time of PD diagnosis and declined over time. Specific T cell subsets involved in responses included CD4+ T cells producing interferon gamma (IFN-γ) and interleukin 4 (IL-4), and both CD4+ and CD8+ T cells producing interleukin 10 (IL-10). T cells producing IL-10 were distinct from T cells producing pro-inflammatory cytokines but were not found to express markers associated with T regulatory (Treg) cells [56]. These studies suggest a pro-inflammatory role in PD progression for T helper 1 (Th1) and Th2 subsets of α-syn-specific CD4+ T cells, as well as a potential regulatory role for a T cell subset that is yet to be identified. Mapping and comparison of the TCR repertoire of α-syn-specific CD4+ T cells in PD patients revealed a repertoire that was specific to each patient. There were no shared TCRs identified [57] and therefore no evidence of a public T cell response in which individuals share identical TCRs in response to the same antigenic epitope [58]. It is possible that α-syn-specific T cells exhibit the same or similar responses in other synucleinopathies such as DLB; however, whether they are recruited into the brain is yet to be established. Despite evidence of T cell co-localisation to α-syn in the DLB brain, it remains unclear whether α-syn plays a fundamental role in T cell reactivity and proliferation in this disease.

It is important to consider the mechanisms by which T cells can infiltrate the brain parenchyma. Immunohistochemistry on PDD meninges revealed T cells expressing the chemokine receptor C-X-C motif chemokine receptor 4 (CXCR4) in response to the presence of its ligand, C-X-C motif chemokine ligand 12 (CXCL12). This was supported by examination of the substantia nigra in 2 DLB and 4 PDD cases, which detected T cells located in the perivascular spaces next to blood vessels expressing CXCL12 [50]. CXCL12 and CXCR4 are known to be involved in the migration of T cells from perivascular spaces into the parenchyma, following their passage through postcapillary venules [59]. In experimental autoimmune encephalomyelitis, T cells that express CXCR4 have been shown to accumulate in perivascular spaces surrounded by endothelial cells expressing CXCL12 [60]. In the presence of inflammation, CXCL12 expression is reduced due to the upregulation of atypical chemokine receptor 3 (ACKR3), a specific scavenger for CXCL12, and results in the migration of T cells out of the perivascular space into the parenchyma [61]. CXCL12 may therefore play a role in limiting the parenchymal infiltration of T cells in the absence of inflammation, although it is unknown if this is replicated in DLB.

In summary, there is evidence of increased parenchymal infiltration of T cells in DLB, particularly near α-syn pathology, suggesting that T cell recruitment is a feature of disease pathophysiology. Predominant infiltration of CD4+ cells may be influenced by CXCL12, although the mechanism and consequences of this remain unclear. Interaction between infiltrating peripheral T cells and microglia, as well as the ability of α-syn to induce inflammatory changes, may play a role in neuroinflammation and neurodegeneration in DLB. However, the precise subtypes of T cells and the brain areas in which they are involved in DLB has been less well-defined. A synthesis of the evidence showing a potential role of T cells contributing to neuroinflammation in DLB is summarised in Figure 1.

### 3.2. T Cells in the CSF Compartment

Examination of CSF in AD and PD subjects has consistently shown increased numbers of activated T cells [62,63]. In contrast, only one study to date has reported on the frequency of T cell populations in the CSF in DLB, but this has provided valuable insight into peripheral neuroinflammatory mechanisms. In this study investigating the CSF in PD–DLB and control subjects, CD4+ T cells in diseased subjects were detected to have a transcriptionally altered immune cell subtype, as defined by higher expression of the *cd69* gene, an early activation marker on T cells, and of the chemokine receptor CXCR4 [50]. Higher levels of CXCL12 were identified in the CSF of PD–PDD subjects associated with neurodegenerative markers, including the neurofilament light chain [50], reflecting neuronal damage [64]. As previously discussed, CXCL12 has been implicated in the infiltration of T cells from perivascular spaces into the parenchyma in DLB. Furthermore, studies examining CSF in subjects with neuroinflammatory diseases have demonstrated positive correlation between levels of CXCL12 and evidence of BBB disruption [65,66]. The presence of CXCL12 in the CSF may therefore play a role in T cell migration across the BBB in PD–PDD, but caution should be applied as to whether this hypothesis applies to DLB.

### 3.3. T Cells in the Blood Compartment

There is growing evidence to suggest alterations in peripheral adaptive immune cell subsets in neurodegenerative diseases, but few studies have examined peripheral T cell subsets in DLB.

In humans, our group examined T cell subsets in the periphery in 27 DLB subjects, 27 AD subjects and 28 healthy controls. We found significantly lower proportions of CD4+ T cells and reduced HLA–DR+ activated B cells in DLB compared to AD. There was also a trend towards increased CD8+ T cells in DLB compared to controls, and a trend towards relative numbers of CD8+ terminal effector cells being increased and naïve CD8+ T cells being reduced in DLB compared to controls, although these results were not statistically significant [67]. Terminal effector cells are known to express markers of senescence. They have limited proliferative capacity but tend to produce a wider range of cytokines following activation [68]. Analysis of peripheral cytokines in this study showed significantly higher interleukin 1 beta (IL-1β) and interleukin 6 (IL-6) in DLB compared to controls [67]. IL-6 induces the development of Th17 cells and inhibits Treg cell differentiation, therefore acting as a pro-inflammatory cytokine in T cells [69]. IL-1β is known to be a potent inducer of IL-6 [70] and, in addition to its presence in the periphery, has been shown to be upregulated in the substantia nigra and frontal cortex of PD–PDD brains [44]. In addition, increased peripheral levels of IL-1β and IL-6 can disrupt BBB integrity [71], providing a mechanism for T cell infiltration into the brain. This study revealed an altered peripheral immune profile in DLB, with reduced proportions of CD4+ T cells and reduced activation of B cells in subjects with DLB, alongside a potential shift from naïve to terminal effector CD8+ T cells, suggesting an immunosenescent profile, as has been reported previously in AD [72]. This immune profile may result in the preferential expression of pro-inflammatory cytokines, contributing to BBB disruption and T cell infiltration into the DLB brain.

## 4. Conclusions

We have provided an overview of current research into the profile of T cells in the brain, CSF, and blood compartment in DLB, and have summarised our main findings in Figure 2.

We have reviewed evidence showing increased recruitment of CD4+ T cells into the human brain parenchyma in DLB, near to α-syn aggregates and Iba1+ microglia. The presence of α-syn appears to play a key role in promoting inflammatory changes through interaction with microglia and T cells, contributing to neuroinflammation and neurodegeneration. Higher expression of pro-inflammatory cytokines, such as IL-17A and IL-1β, in DLB brain tissue supports that infiltrating T cells may have pathogenic rather than neuroprotective properties. It is important to consider that post-mortem studies largely examine later-stage disease and results may not represent earlier disease mechanisms. Post-mortem brain tissue from subjects in early stages of DLB would be highly valuable to examine, although the availability of relevant brain tissue will limit the power of such studies. Few studies have explored T cell profiles in DLB cases where there is known concomitant AD pathology, and further work is required to clarify T cell subsets in the DLB brain. The interaction between α-syn, AD pathology, microglia, and T cells is therefore a key area for future research.

Enhanced activation of CD4 T cells has been detected in the CSF of PD–DLB subjects, with increased CSF CXCL12 associated with neurodegeneration. The CXCR4–CXCL12 signalling pathway may recruit T cells into the brain and has the potential to guide new therapeutic options, such as CXCR4 antagonists. However, this does rely on the assumption that the pathophysiology of DLB is similar to that in PD and PDD, which requires further validation.

The profile of T cells in the blood of DLB subjects shows reduced CD4+ T cells and a trend towards increased CD8+ T cells. This suggests impaired proliferation and/or reduced activation of CD4+ T cells in DLB and differs from the changes associated with normal ageing. Studies examining peripheral markers of senescence or exhaustion in DLB would provide further information on the extent of T cell differentiation. Longitudinal analysis of blood and CSF in DLB from prodromal to terminal stages would also enhance our understanding of how the adaptive immune response changes with disease progression.

We advocate further research to determine the precise role of T cells in the pathophysiology of DLB. A greater understanding of the pathogenic and neuroprotective properties of different T cell subtypes in DLB may help guide the development of new therapeutic strategies.

## Figures and Tables

**Figure 1 cells-12-02283-f001:**
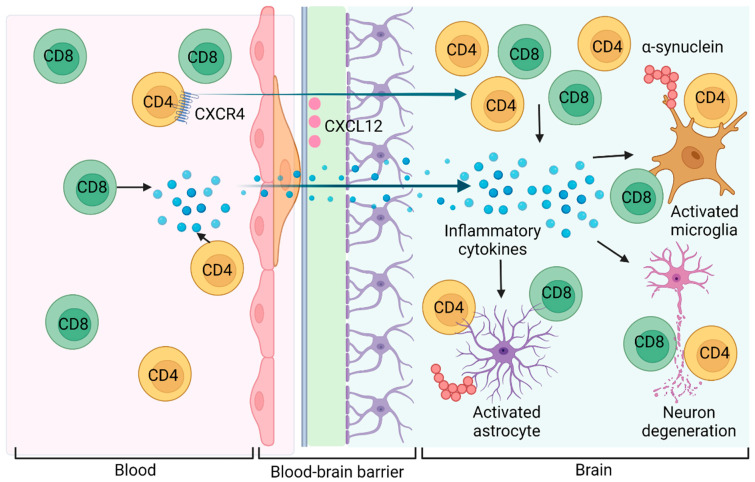
The potential role of T cells contributing to neuroinflammation in DLB. Extracellular aggregates of α-syn activate microglia and astrocytes, inducing an inflammatory response involving T cell activation and infiltration into the parenchyma. T cells expressing CXCR4 localise to CXCL12, which may play a role in the migration of T cells across the BBB. Created by Biorender.com (accessed 31 July 2023).

**Figure 2 cells-12-02283-f002:**
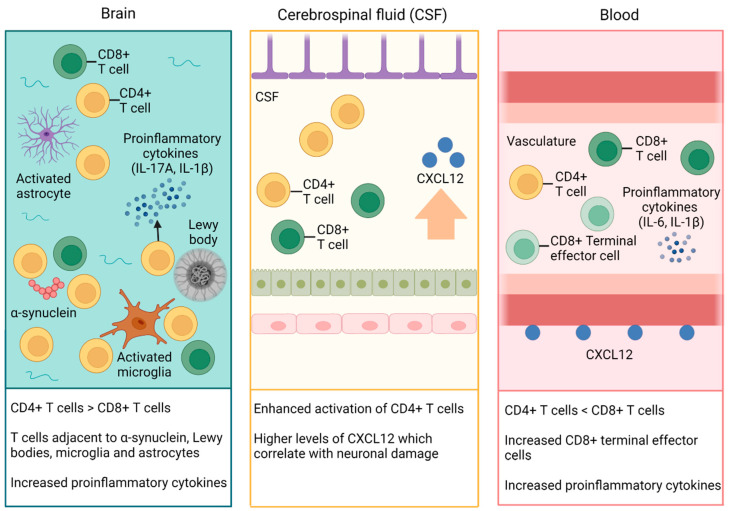
Alterations in the profile of T cells in the brain, CSF, and blood in DLB. Increased numbers of T cells adjacent to α-syn and microglia in the brain, enhanced CD4+ T cell activation in the CSF, and reduced numbers of CD4+ T cells in the blood have been demonstrated in DLB. The presence of CXCL12 in perivascular spaces and increased levels of CXCL12 in the CSF may contribute to disruption of the BBB and infiltration of T cells into the parenchyma. Created with Biorender.com (last accessed 31 July 2023).

**Table 1 cells-12-02283-t001:** Characteristics of CD4+ T cell subsets.

Subset	Effector Cytokines Produced	Functions
Th1	IFN-γ, Lymphotoxin, TNFα	Cell-mediated immunity Delayed-type hypersensitivity responses Clearance of intracellular pathogens
Th2	IL-4, IL-5, IL-13, IL-10	Humoral immunity Clearance of extracellular worms and bacteria B cell switching to IgE Allergic responses
Th9	IL-9, IL-10	Protection against parasitic worms
Th17	IL-17, IL-17F, IL-6, IL-22, TNFα, IL-10	Protection of mucosal surfaces Recruitment of neutrophils Clearance of *Mycobacterium tuberculosis* and *Klebsiella pneumonia*
Th22	IL-22, IL-13, FGF, CCL15, CCL17, TNFα	Mucosal immunity Prevention of microbial translocation across epithelial surfaces Promotes wound repair
Th25	IL-25, IL-4, IL-5, IL-13	Mucosal immunity Stimulates non-lymphoid cells to produce IL-4 Limits Th1- and Th7-induced inflammation
ThFH	IL-21, TNFRSF4, ICOS	Helps B cells to produce high affinity antibodies Triggers the formation and maintenance of germinal centres
Treg	IL-10, TGFβ	Suppression of existing immune responses Maintains tolerance against autoimmunity

Th (T helper), ThFH (T follicular helper cell), Treg (T regulatory cell), IFN-γ (Interferon-gamma), TNFα (Tumour necrosis factor-alpha) IL (Interleukin), FGF (Fibroblast growth factor), CCL (Chemokine ligand), TNFRSF4 (Tumour necrosis factor receptor superfamily member 4), ICOS (Inducible T cell costimulator), TGFβ (Transforming growth factor-beta). Adapted from Caza et al. 2015 [29].

**Table 2 cells-12-02283-t002:** Characteristics of CD8+ T cell subsets.

Subset	Effector Cytokines Produced	Functions
Tc1	IFN-γ, TNFα	Clearance of intracellular pathogens and tumours
Tc2	IL-4, IL-5, IL-13	Propagation of Th2-mediated allergy
Tc9	IL-9, IL-10	Inhibition of CD4+ T cell-mediated colitis Propagation of Th2-mediated allergy Anti-tumour response
Tc17	IL-17, IL-21	Promotes autoimmune responses Immunity to viral infections Anti-tumour response
Tc22	IL-22, IL-2, TNFα	Anti-tumour response
Treg	TGFβ, IL-10	Regulation of T cell-mediated responses

Tc (T cytotoxic), Treg (T regulatory cell), IFN-γ (Interferon-gamma), TNFα (Tumour necrosis factor-alpha) IL (Interleukin), TGFβ (Transforming growth factor-beta). Adapted from Mittrucker et al. 2014 [30].

## Data Availability

No new data were created or analysed in this study. Data sharing is not applicable to this article.

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
