# Peer review of "T Lymphocytes and Their Potential Role in Dementia with Lewy Bodies"

_cells, 2023, doi:10.3390/cells12182283_

Round 1
Reviewer 1 Report
Amin et al. provide a well-written and comprehensive review of T cell changes in Dementia with Lewy bodies. Although several reviews on T cell changes in Alzheimer's disease have been recently published, the focus on dementia with Lewy bodies is significant and timely. This current review highlights the potential involvement of T cells in multiple neurodegenerative diseases, expanding on the rapidly evolving field of T cells in Alzheimer's disease. Overall, this review is excellent, and I only found a few areas that need clarification.
One minor error on Lines 97-99: "The adaptive immune system comprises T and B cells, antigen-presenting cells (APCs, such as macrophages), and antibodies." As written, the sentence suggests that macrophages are not part of the adaptive immune system, which is inaccurate. Please revise.
My second comment pertains to the introduction and Tables 1 and 2. The introduction shows a notable comparison between DLB, AD, and PDD. However, Tables 1 and 2 only discuss DLB and PDD. The author should consider including AD in the tables or clarifying in the introduction why only DLB and PDD are discussed in Tables 1 and 2.
